# Clinical, anamnestic, and sociodemographic predictors of positive SARS-CoV-2 testing in children: A cross sectional study in a tertiary hospital in Italy

Benedetta Armocida[1], Giulia Zamagni[1], Elena Magni[1], Lorenzo Monasta[1], Manola Comar[2,3], Nunzia Zanotta[2], Carolina Cason[2], Giorgia Argentini[4], Marianela Urriza[4], Andrea Cassone[5], Fulvia Vascotto[5], Roberto Buzzetti[6], Egidio Barbi[3,7], Massimo Del Pin[8], Paola Pani[1], Alessandra Knowles[1], Claudia Carletti[1], Federica Concina[1], Mariarosa Milinco[1], Luca Ronfani[1]*

1 Clinical Epidemiology and Public Health Research Unit, Institute for Maternal and Child Health—IRCCS "Burlo Garofolo", Trieste, Italy, 2 Unit of Advanced Microbiology Diagnosis and Translational Research, Institute for Maternal and Child Health—IRCCS "Burlo Garofolo", Trieste, Italy, 3 University of Trieste, Trieste, Italy, 4 Medical Direction, Institute for Maternal and Child Health—IRCCS "Burlo Garofolo", Trieste, Italy, 5 Health Professions Direction, Institute for Maternal and Child Health—IRCCS "Burlo Garofolo", Trieste, Italy, 6 Epidemiologist, Bergamo, Italy, 7 Department of Pediatrics, Institute for Maternal and Child Health—IRCCS "Burlo Garofolo", Trieste, Italy, 8 Department of Medicine, University of Udine, Udine, Italy

* luca.ronfani@burlo.trieste.it

## Abstract

### Objectives

We aimed to identify clinical, anamnestic, and sociodemographic characteristics associated with a positive swab for SARS-CoV2, and to provide a predictive score to identify at risk population in children aged 2–14 years attending school and tested for clinical symptoms of COVID-19.

### Design

Cross sectional study.

### Setting

Outpatient clinic of the IRCCS Burlo Garofolo, a maternal and child health tertiary care hospital and research centre in Italy.

### Data collection and analysis

Data were collected through a predefined form, filled out by parents, and gathered information on sociodemographic characteristics, and specific symptoms, which were analysed to determine their association with a positive SARS-CoV-2 swab. The regression coefficients of the variables included in the multivariate analysis were further used in the calculation of a predictive score of the positive or negative test.

**Data Availability Statement:** All relevant data are within the paper and its Supporting Information files.

**Funding:** The authors received no specific funding for this work.

**Competing interests:** The authors have declared that no competing interests exist.

## Results

Between September 20th and December 23rd 2020, from 1484 children included in the study, 127 (8.6%) tested positive. In the multivariate analysis, the variables retained by the model were the presence of contact with a cohabiting, non-cohabiting or unspecified symptomatic case (respectively OR 37.2, 95% CI 20.1–68.7; 5.1, 95% CI 2.7–9.6; 15.6, 95% CI 7.3–33.2); female sex (OR 1.49, 95% CI 1.0–2.3); age (6–10 years old: OR 3.2, 95% CI 1.7–6.1 p<0.001; >10 years old: OR 4.8, 95% CI 2.7–8.8 p<0.001); fever (OR 3.9, 95% CI 2.3–6.4); chills (OR 1.9, 95% CI 1.1–3.3); headache (OR 1.45, 95% CI 0.9–2.4); ageusia (OR 1.3, 95% CI 0.5–4.0); sore throat (OR 0.48, 95% CI 0.3–0.8); earache (OR 0.4, 95% CI 0.1–1.3); rhinorrhoea (OR 0.8, 95% CI 0.5–1.3); and diarrhoea (OR 0.52, 95% CI 0.2–1.1). The predictive score based on these variables generated 93% sensitivity and 99% negative predictive value.

## Conclusions

The timely identification of SARS-CoV2 cases among children is useful to reduce the dissemination of the disease and its related burden. The predictive score may be adopted in a public health perspective to rapidly identify at risk children.

## Introduction

Coronavirus-19 disease (COVID-19) predominantly affects adults, whereas children are mostly asymptomatic or affected by less severe clinical pictures, with symptoms like those of other acute respiratory illnesses (fever, cough, runny nose, nasal congestion, fatigue) [1–3]. Evidence available on the differential diagnosis between SARS-CoV-2 and coexisting seasonal infections are limited [4]. At present, most of the studies described symptoms in SARS-CoV-2 positive subjects, while a very limited number of studies tried to characterize clinical and epidemiological risk factors able to predict the positivity for SARS-CoV-2 [5–9]. Health care professionals and policy-makers could benefit from knowing which presenting characteristics and symptoms are most likely to be associated with SARS-CoV-2 infection. In Italy, the Prime Minister's Decree of September 7, 2020, specified operational measures aimed at managing cases of SARS-CoV-2 on school grounds and contains a list of symptoms suggestive of the disease, useful to identifying children to be placed in isolation and undergo laboratory testing [10]. Unfortunately, this list included a wide range of non-specific symptoms and consequently, given the high frequency of upper respiratory and gastrointestinal infections linked to viruses other than SARS-CoV-2, many children attending school and educational services are at risk of undergoing unnecessary testing. Knowing which clinical, anamnestic, and sociodemographic characteristics are more likely to be associated with SARS-CoV-2 infection may help to target testing criteria for children.

The aim of this study was to identify clinical, anamnestic, and sociodemographic characteristics associated with a positive swab for SARS-CoV2 in a population of children between 2 and 14 years of age attending school and educational services and tested for clinical symptoms of COVID-19, and to provide a predictive score to identify children at risk.

## Methods

### Study design and setting

This was a cross-sectional study, reported using the Standards for Strengthening the Reporting of Observational Studies in Epidemiology (STROBE) [11] (S1 Table).

The study was conducted between 20th September and 23rd December 2020 at the outpatient clinic of the IRCCS Burlo Garofolo, a maternal and child health tertiary care hospital and research centre located in Trieste, Italy, and covering the whole area of the Trieste Province (about 230.000 inhabitants). In the study period the Institute laboratory was the only one processing children's swab in the Trieste area.

## Study population

In this study were enrolled children between 2–14 years of age, attending school or other educational services. Children with clinical symptoms undergoing reverse transcription-polymerase chain reaction (RT-PCR) nasopharyngeal swab to detect SARS-CoV-2 at the outpatient clinic, were included in the study, while asymptomatic children tested for contact-tracing or as follow ups were excluded.

## Data collection

Data were collected through a predefined anamnestic form, independently filled out by the families when the RT-PCR testing was performed. The first section of the form collected information on sociodemographic characteristics—age, sex, citizenship, the health professionals, or facilities requesting the test, any returns from abroad with indication of the country in question, the school attended, the presence of positive contacts cohabiting or not cohabiting and the presence of health workers in the family, with specific indication of their place of work. The second part of the form collected information on specific symptoms that led to the swab being administered, such as fever>37.5°, fatigue, bone pain, chills, colds, cough, sore throat, earache, wheezing, shortness of breath, chest pain, headache, stomach pain, nausea/vomiting, diarrhoea, anosmia, ageusia, conjunctivitis.

## Outcomes

The primary outcome of the study was the positivity to molecular nasopharyngeal swab.

## Laboratory methods

Total RNA was extracted from nasopharyngeal swab starting from 200 μl in a final elution volume of 50 μl, using the Maxwell CSC Instrument (Promega Srl, Italy) and following the manufacturer's instructions.

SARS-CoV-2 detection was performed on the CFX96TM Real-Time PCR Detection System (Bio-Rad, California, USA), using the NeoPlexTM COVID-19 Detection Kit (Genematrix, Seongnam, Kyonggi-do, South Korea) targeting the viral N and RdRp genes and and housekeeping gene of β-actin as internal control, following the manufacturer's instructions. The sensitivity of the test is of 8 viral copies/ul of sample. Finally, for each swab the quantity and quality of the material taken was checked to control that it was sufficient to highlight up to 5 copies of the virus, and in case of a doubtful result the patient was asked to re-perform the swab.

## Data analysis

Data were reported as number and percentage for categorical variables and as mean and standard deviations for continuous variables. Between groups differences (subjects positive vs negative to RT-PCR test) were evaluated with the chi-square test (or the Fisher exact test, when appropriate) for categorical variables and with the Student's t-test for continuous variables.

The study population was subdivided into three age groups based on the school divisions presented in Italy (<6 years old; 6–10 years old; >10 years old).

A multivariate logistic model was estimated with variables selected through the Lasso penalty method, to identify variables that were strongly associated with a positive test for SARS-CoV-2. The Lasso procedure was performed using cross-validation with 5 folds. The model with lambda minimizing the Mean Squared Prediction Error (MSPE) was selected, in our case lambda = 6. The regression coefficients of the variables retained by the Lasso multivariate analysis were used in the calculation of a predictive score of the positive or negative test. The score for a single individual corresponds to the sum of the coefficients of the characteristics and symptoms presented, rounded to the nearest 0.5. Receiver operating characteristic (ROC) curves were created from the individual scores, to assess predictor performance after selecting relevant classifiers based on the regression model, and a cut-off was identified capable of maximizing sensitivity without causing an excessive loss of specificity. Area under the ROC curve (AUC) analysis was performed to assess the ability of classifiers to discriminate COVID-19 positive subjects from COVID-19–negative subjects. Statistical significance was set at p-value <0.05 for all analyses, even if for the development of the predictive score the selection of variables was based on Lasso penalisation criteria. Data were analysed with StataCorp. 2019. *Stata Statistical Software*: *Release 16*. College Station, TX: StataCorp LLC.

## Ethical considerations

The study was approved by the Institutional Review Board of the Institute for Maternal and Child Health IRCCS Burlo Garofolo (IRB_BURLO 03/2020, 20.05.2020). Data were anonymised during data entry once the swab results was received. Data were analysed and reported only in aggregate form.

## Results

Between 20th September and 23rd December 2020, 1484 children were included in the study of which 127 (8.6%) tested positive at RT-PCR test (Fig 1).

The socio-demographic and exposure characteristics of the children by group are described in Table 1. The distribution by age group was significantly different between positive and negative (p<0.001). Specifically, 55% of children in the positive group were in the age group ≥ 10 years compared to 32% in the negative group. Considering the subgroups' population division, the emerged differences among ages groups reflected also the school attended. Contact with a symptomatic individual was reported for 63% of positives vs 16% of negatives (p<0.001). No statistically significant differences emerged between the two groups on regards of sex, citizenship, and presence of a health worker in the family.

The symptoms presented are described in Table 2. Fever was prevalent in both groups, significantly more in SARS-CoV-2 positive children (63% vs 43%, p<0.001). Other common symptoms significantly different between positive subjects and negative were headache (51% vs 30% p<0.001), chills (29% vs 14% p<0.001), and rhinorrhoea (46% vs 62% p<0.001).

In the Lasso multivariate analysis, socio-demographic factors retained by the model were the presence of contact with a cohabiting, non-cohabiting or unspecified symptomatic case, female sex, and age (6–10 years old, and >10 years old) (Table 3). Among clinical symptoms, the presence of fever, chills, headache, and ageusia were retained by the model and associated with an increased risk of testing positive, while sore throat, earache, rhinorrhoea, and diarrhoea with a reduced risk (Table 3).

For each subject, the sum of the coefficients assumed a value between -2 and +8.5 (Table 4).

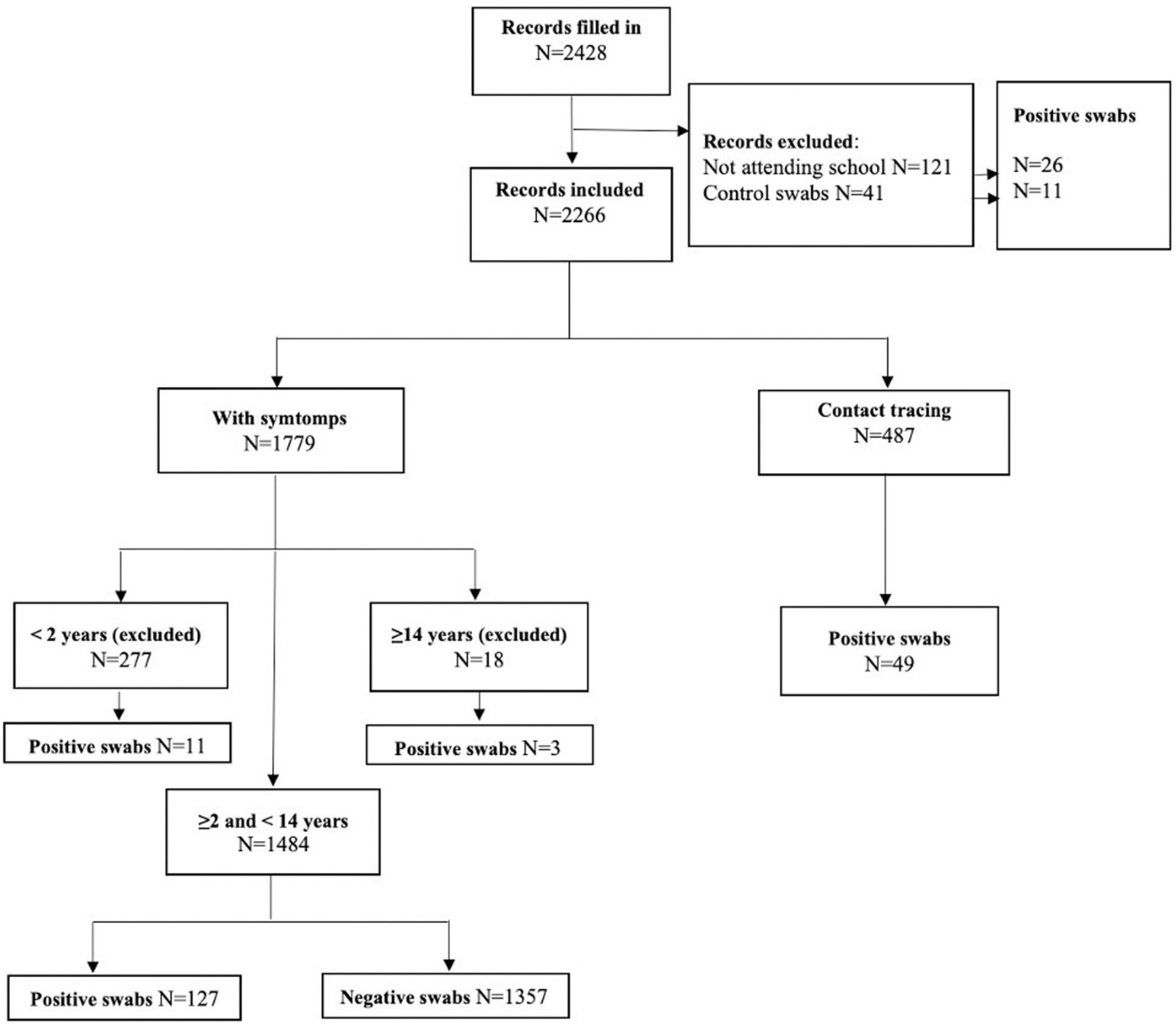

**Fig 1. Flow diagram.**

The ROC curve constructed from individual scores had an area under the curve of 0.88 (95% CI 0.86–0.92) (Fig 2). A score cut-off value >2 maximised the sum of sensitivity and specificity and, applied to our sample, generated the 93% sensitivity with 65% specificity, 20% positive predictive value and 99% negative predictive value.

## Discussion

This study showed that rhinorrhoea, sore throat, and diarrhoea were associated with a reduced probability of testing positive to SARS-CoV-2, while confirmed an increased risk in children presenting fever, chills, ageusia and headache. To our knowledge, this is the first study to

**Table 1. Socio-demographic characteristics and epidemiological profile of children aged 2–14 years old.**

|  | Positive swab N = 127 | Negative swab N = 1357 | P value | Total N = 1484 |
|---|---|---|---|---|
| **Age, n (%)** |  |  | <0.001 |  |
| <6 years | 27 (21.3) | 589 (43.4) |  | 616 (41.5) |
| 6–10 years | 30 (23.6) | 330 (24.3) |  | 360 (24.3) |
| >10 years | 70 (55.1) | 438 (32.3) |  | 508 (34.2) |
| **Sex, n (%)** |  |  | 0.078 |  |
| Males | 54 (42.5) | 693 (51.1) |  | 747 (50.3) |
| Females | 73 (57.5) | 664 (48.9) |  | 737 (49.7) |
| **Italian citizen, n (%)** | 121 (96.8) | 1259 (93.6) | 0.176 | 1380 (93.9) |
| **School attended, n (%)** |  |  | <0.001 |  |
| Nursey school | 4 (3.1) | 138 (10.2) |  | 142 (9.5) |
| Kindergarden | 23 (18.1) | 432 (31.8) |  | 455 (30.7) |
| Primary school | 42 (33.1) | 436 (32.1) |  | 478 (32.2) |
| Middle school | 57 (44.9) | 339 (25.0) |  | 396 (26.7) |
| High school | 1 (0.8) | 12 (0.9) |  | 13 (0.9) |
| **Contact with symptomatic patients, n (%)** |  |  | <0.001 |  |
| No/I don't know | 47 (37.0) | 1144 (84.3) |  | 1191 (80.2) |
| Yes, cohabitants | 45 (35.4) | 57 (4.2) |  | 102 (6.9) |
| Yes, no cohabitant | 19 (15.0) | 116 (8.6) |  | 135 (9.1) |
| Yes, not specified | 16 (12.6) | 40 (2.9) |  | 56 (3.8) |
| **Presence of hws in the family, n (%)** | 13 (10.3) | 146 (10.9) | 1.000 | 159 (10.8) |

Abbreviation: hws = health workers.

provide a predictive score for the paediatric population to identify children at risk to be positive to the SARS-CoV-2 test.

**Table 2. Clinical presentations of children aged 2–14 years old.**

|  | Positive N = 127 | Negative N = 1357 | OR | 95% CI | p-value |
|---|---|---|---|---|---|
| Fever >37.5, n (%) | 80 (63.0) | 588 (43.3) | 2.22 | 1.53–3.24 | <0.001 |
| Chills, n (%) | 37 (29.1) | 187 (13.8) | 2.57 | 1.70–3.88 | <0.001 |
| Fatigue, n (%) | 48 (37.8) | 361 (26.6) | 1.67 | 1.15–2.44 | 0.008 |
| Muscle pain, n (%) | 25 (19.8) | 152 (11.2) | 1.96 | 1.23–3.13 | 0.005 |
| Sore throat, n (%) | 31 (24.4) | 507 (37.4) | 0.54 | 0.36–0.82 | 0.004 |
| Cough, n (%) | 49 (38.6) | 657 (48.4) | 0.67 | 0.46–0.97 | 0.035 |
| Rhinorrhoea, n (%) | 58 (45.7) | 847 (62.4) | 0.50 | 0.35–0.73 | <0.001 |
| Earache, n (%) | 4 (3.1) | 77 (5.7) | 0.54 | 0.19–1.50 | 0.238 |
| Respiratory distress, n (%) | 5 (3.9) | 40 (2.9) | 1.36 | 0.53–3.51 | 0.526 |
| Wheezing, n (%) | 1 (0.8) | 33 (2.4) | 0.32 | 0.04–2.35 | 0.262 |
| Thoracic pain, n (%) | 5 (3.9) | 26 (1.9) | 2.12 | 0.80–5.61 | 0.132 |
| Headache, n (%) | 65 (51.2) | 412 (30.4) | 2.4 | 1.67–3.47 | <0.001 |
| Stomach-ache, n (%) | 23 (18.1) | 244 (18.0) | 1.02 | 0.63–1.63 | 0.945 |
| Nausea/vomiting, n (%) | 14 (11.0) | 165 (12.2) | 0.89 | 0.50–1.60 | 0.705 |
| Diarrhoea, n (%) | 10 (7.9) | 152 (11.2) | 0.68 | 0.35–1.33 | 0.263 |
| Ageusia, n (%) | 8 (6.3) | 32 (2.4) | 2.78 | 1.25–6.17 | 0.012 |
| Anosmia, n (%) | 9 (7.1) | 45 (3.3) | 2.22 | 1.06–4.66 | 0.034 |
| Conjunctivitis, n (%) | 5 (3.9) | 21 (1.5) | 2.61 | 0.97–7.04 | 0.058 |

**Table 3. Results of the Lasso multivariate logistic regression analysis for children of the 2–14 years group.**

|  | OR | 95% CI | p-value |
|---|---|---|---|
| Contact (ref No contact) |  |  |  |
| with symptomatic patients' cohabitants | 37.2 | 20.1–68.7 | <0.001 |
| with symptomatic patients no cohabitants | 5.1 | 2.7–9.6 | <0.001 |
| with symptomatic patients no specified | 15.6 | 7.3–33.2 | <0.001 |
| Age (ref 2–6 years) |  |  |  |
| 6–10 years | 3.2 | 1.7–6.1 | <0.001 |
| >10 years | 4.8 | 2.7–8.8 | <0.001 |
| Fever | 3.9 | 2.3–6.4 | <0.001 |
| Chills | 1.9 | 1.1–3.3 | 0.020 |
| Female sex | 1.49 | 1.0–2.3 | 0.073 |
| Headache | 1.45 | 0.9–2.4 | 0.144 |
| Ageusia | 1.3 | 0.5–4.0 | 0.589 |
| Earache | 0.4 | 0.1–1.3 | 0.137 |
| Diarrhoea | 0.52 | 0.2–1.1 | 0.091 |
| Sore throat | 0.48 | 0.3–0.8 | 0.005 |
| Rhinorrhoea | 0.8 | 0.5–1.3 | 0.449 |

In our study, a minority of children tested were positive to SARS-CoV-2. This proved that children are less affected by COVID-19 than adults, which might have also determined a reduced perform of diagnostic testing, with a consequent underestimation of the real numbers of infected children [12, 13].

A study based in the Unites States (US) reported children under 6 years being most frequently tested, while older children having a higher frequency to be positive [14]. These findings were in line with our study, which indicated an association between older age (6–10 years old: [OR 3.2, 95% CI 1.7–6.1]; >10 years old: [OR 4.8, 95% CI 2.7–8.8]) and the odds of a

**Table 4. Predictive score resulted from the calculation of regression coefficients of the variables retained by the Lasso multivariate analysis.**

|  | Score |
|---|---|
| No contact with symptomatic patients (ref.) | 0 |
| Contact with symptomatic patients' cohabitant | 3.5 |
| Contact with symptomatic patients no cohabitant | 1.5 |
| Contact with symptomatic patients no specified | 2.5 |
| Age 2–6 years (ref.) | 0 |
| Age 6–10 years | 1 |
| Age >10 years | 1.5 |
| Fever | 1.5 |
| Chills | 0.5 |
| Sex male (ref.) | 0 |
| Sex female | 0.5 |
| Headache | 0.5 |
| Ageusia | 0.5 |
| Earache | -1 |
| Sore throat | -0.5 |
| Diarrhoea | -0.5 |
| Rhinorrhoea | 0 |

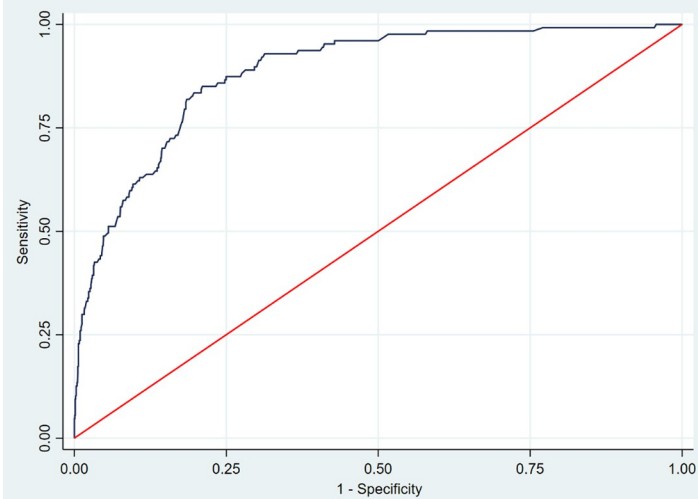

**Fig 2. ROC curve of the Lasso multivariate logistic model.**

positive test result [7, 14–16]. Our analyses also reported a higher probability to test positive in children with contact with a symptomatic person (cohabitants: OR 37.2, 95% CI 20.1–68.7; no-cohabitants: OR 5.1, 95% CI 2.7–9.6; not specified: OR 15.6, 95% CI 7.3–33.2), highlighting the importance of prioritising the test for older children and those with a history of exposure. Differently from other studies, we found an increased risk of a positive SARS-CoV-2 test in females (OR 1.49, 95%CI 1.0–2.3), and no association with citizenship (Italian vs foreign). In line with a large US report on 69,703 paediatric cases of COVID-19, systematic reviews and studies conducted in Italy, fever and cough were the most common symptoms [12, 17–20]. The symptoms described in our study were similar to those reported for school-aged children in a study carried out in an Italian outpatient setting [8]. Our results not only showed a more probable positive test in children presenting fever, but we also found chills, ageusia and headache related to a higher risk of a positive test, while rhinorrhoea, sore throat, and diarrhoea as reduced risks. These findings confirmed how challenging it is to differentiate, only through clinical symptoms, SARS-CoV-2 positive cases from other viral diseases which can affect children and have similar clinical presentation, with the only exception of ageusia, which is quite a specific sign. In a recent study anosmia-ageusia have been reported as the most robustly associated symptom with SARS-CoV-2 test positivity [21], and in line with our results were overall more common among individuals reporting positive test results than among those reporting negative test results.

A very limited number of children presented wheezing (34/1484, 2.2%), particularly in SARS-CoV-2 positive cases (0.8% vs 2.4% in negative cases, Table 2). This finding might have been the result of a misreporting of symptoms by parents, even though all cases were referred for swab by the family paediatrician, or the changing in the seasonality of winter paediatric respiratory infections as a consequence of social distancing measures and mask wearing. Preliminary data from the area of Trieste showed that during the last winter season, from September 2020 to February 2021, no influenza A or B nor respiratory syncytial virus (RSV) were detected, while rhinovirus, adenoviruses, other coronaviruses, and SARS-CoV-2 were the most common [22]. Furthermore, in line with other studies [23], our data confirm that SARS-CoV-2 did not seem to induce asthma exacerbations, which, instead, have been previously observed in both adults and children with viral upper respiratory tract infections [24].

The study described factors associated with the odds of a positive laboratory-confirmed SARS-CoV-2 infection in paediatric patients presenting symptoms. When accounting for the sensitivity, specificity, positive predictive value, and negative predictive value of different combinations of socio-demographic characteristics and symptoms, we found a very sensitive test with a good negative predictive value (respectively 93% and 99%), which minimising the false negatives could be considered a good screening test. Individuals with a risk score >2 should be prioritised for RT-PCR testing. The predictive score provided a non-invasive tool to identify children at higher risk of SARS-CoV-2 infection, and its use may substantially strengthen the effectiveness of the testing, tracing, and isolation approach on which countries' national responses are based [25].

This study presented several limitations. Our analyses are based on a single centred tertiary hospital, and as families were directly providing information, some symptoms might have been misinterpreted. Furthermore, RT-PCR testing presents limits itself. Although the use of this testing is the gold standard currently used to diagnose COVID-19, a high percentage of false negative cases has been reported in the literature, that might be attributable to the low viral loads, particularly in asymptomatic or mild symptomatic patients that might transmit the disease as well [18, 26]. Moreover, the accuracy of the RT-PCR testing highly depends on the period when the test is performed. We also acknowledge the possible limit in performing only one single test, which, without a further systematic follow-up, might have not detected a child positive to SARS-CoV-2, who might have been included in the negative control group. However, we considered this possibility extremely small, as our Institute was the only hub performing COVID-19 swab in the Trieste area, hence, in case of persistent or worsening symptoms, the access of these children for a second swab would have been performed within our Institute and this didn't happen from our available data. Finally, new virus variants might eventually determine different clinical presentations, with variation of symptoms and their intensity.

Even though the results of this study may be useful to health professionals to rapidly identify children at increased risk of COVID-19, and consequently take all necessary containment measures for reducing the dissemination of the disease, it might also help directing national COVID-19 response to control transmission and to reduce the disease burden. This information might be also relevant in orienting choices in specific settings, such as those with limited resources. Further prospective studies, with a larger population, should be conducted to demonstrate the utility of this score.

## Conclusion

This study identified the factors associated with a higher probability of resulting positive to the RT-PCR test and provided a predictive score which may be adopted in a public health perspective to increase the detection rate of children at risk of infection. In a clinical perspective it showed that rhinorrhoea, sore throat, and diarrhoea were all associated with a reduced probability of testing positive.

## Supporting information

**S1 Table. STROBE statement—checklist of items that should be included in reports of cross-sectional studies.**
(DOCX)

## Author Contributions

**Conceptualization:** Luca Ronfani.

**Data curation:** Manola Comar, Nunzia Zanotta, Carolina Cason, Giorgia Argentini, Maria-nela Urriza, Fulvia Vascotto, Paola Pani, Alessandra Knowles, Claudia Carletti, Federica Concina, Mariarosa Milinco.

**Formal analysis:** Giulia Zamagni, Elena Magni, Lorenzo Monasta, Roberto Buzzetti, Luca Ronfani.

**Investigation:** Giorgia Argentini, Andrea Cassone, Fulvia Vascotto, Massimo Del Pin, Paola Pani, Alessandra Knowles, Claudia Carletti, Federica Concina, Mariarosa Milinco.

**Methodology:** Lorenzo Monasta, Manola Comar, Giorgia Argentini, Paola Pani, Alessandra Knowles, Claudia Carletti, Federica Concina, Mariarosa Milinco, Luca Ronfani.

**Project administration:** Paola Pani, Alessandra Knowles, Claudia Carletti, Federica Concina, Mariarosa Milinco, Luca Ronfani.

**Supervision:** Luca Ronfani.

**Validation:** Benedetta Armocida, Egidio Barbi, Luca Ronfani.

**Writing – original draft:** Benedetta Armocida, Giulia Zamagni, Luca Ronfani.

**Writing – review & editing:** Benedetta Armocida, Giulia Zamagni, Elena Magni, Lorenzo Monasta, Manola Comar, Nunzia Zanotta, Carolina Cason, Giorgia Argentini, Marianela Urriza, Andrea Cassone, Fulvia Vascotto, Roberto Buzzetti, Egidio Barbi, Massimo Del Pin, Paola Pani, Alessandra Knowles, Claudia Carletti, Federica Concina, Mariarosa Milinco, Luca Ronfani.

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
