## [Decision Letter · Decision Letter 0]

18 Aug 2021

PONE-D-21-24181

Clinical, anamnestic, and sociodemographic predictors of positive SARS-CoV-2 testing in children: a cross sectional study in a tertiary hospital in Italy

PLOS ONE

Dear Dr. Ronfani,

Thank you for submitting your manuscript to PLOS ONE. After careful consideration, we feel that it has merit but does not fully meet PLOS ONE’s publication criteria as it currently stands. Therefore, we invite you to submit a revised version of the manuscript that addresses the points raised during the review process.

Please address the issues and revise accordingly.

We look forward to receiving your revised manuscript.

Kind regards,

Academic Editor

PLOS ONE

Journal Requirements:

Reviewers' comments:

Reviewer's Responses to Questions

**Comments to the Author**

1. Is the manuscript technically sound, and do the data support the conclusions?

Reviewer #1: Yes

Reviewer #2: Partly

2. Has the statistical analysis been performed appropriately and rigorously? 

Reviewer #1: Yes

Reviewer #2: Yes

3. Have the authors made all data underlying the findings in their manuscript fully available?

Reviewer #1: No

Reviewer #2: Yes

4. Is the manuscript presented in an intelligible fashion and written in standard English?

Reviewer #1: Yes

Reviewer #2: Yes

5. Review Comments to the Author

Reviewer #1: The study is well written and the analyses well performed.

However, there is a significant basic mistake that does not allow to give much significance to this study.

In particular, a single swab performed in the emergency department without information on what happened after discharge has really no sense to me. How can you exclude that you had a false positive result?

For example, more than 200 children from the "negative" group had known contacts. Also, a number of children from the negative group had anosmia and ageusia, which you itself define as a predictive symtpoms. Importantly, these symtpoms are very unusual in other diseases.

As a proof that the study, although well written and analysed, has low significance, you define that a very common and usual presentation of children in the ED can be predictive of covid. However, we know that so many children can have mild or asymptomatic disease.

I am very very sorry to see that you did a huge effort in this analyses, but really the hypothesis behind the design of this study has a very doubtful significance. You should use these data to make a new one, doing f-up swabs as a minimum, or interview of parents to understand if further swabs have been done and the results to see how many of them are true negative. For example, a preprint from UK recenlty found that 1 out of 3 students had IgG positive (!!) confirming that the virus circulates much more than expected. And I am quite sure that among the negative, particularly those with known conttacts and with anosmia/ageusia, you missed several cases.

Reviewer #2: In the manuscript authored by Benedetta Armocida et al, authors summarize clinical, anamnestic, and sociodemographic characteristics of children affected or not by Sars Cov-2 in a frame period between September 20th and December 23rd 2020.

A major concearn is that authors provided a study that is more descriptive then predictive, as stated from authors. This is a critical point. No follow up is present. ROC curve is computed between 127 positive vs 1357 negative. Are these groups comparable?

It is not clear to the reader if authors suggest NOT to perform Sars-Cov2 diagnostic test in children presenting rhinorrhoea, sore throat,or diarrhoea.

6. PLOS authors have the option to publish the peer review history of their article (what does this mean?). If published, this will include your full peer review and any attached files.

Reviewer #1: No

Reviewer #2: No

---

## [Author Response · Author response to Decision Letter 0]

26 Oct 2021

PONE-D-21-24181

Clinical, anamnestic, and sociodemographic predictors of positive SARS-CoV-2 testing in children: a cross sectional study in a tertiary hospital in Italy

Review Comments to the Author

Reviewer #1: The study is well written and the analyses well performed. 

***Thank you for your appreciation.

However, there is a significant basic mistake that does not allow to give much significance to this study.

In particular, a single swab performed in the emergency department without information on what happened after discharge has really no sense to me. 

***Thank you for your comment. At that time, the general process after a positive swab envisaged the activation of quarantine and a swab at two weeks by the Department of Prevention. Thus, a follow up (control) swab was always performed, but these follow up swabs could have been performed in other health services, not necessary in our Institute. Moreover, the aim of this study was to identify clinical, anamnestic, and sociodemographic characteristics associated with a positive swab for SARS-CoV-2 in a population of children, hence we aimed at investigating a real life, pragmatic approach in order to try to add some clinical hints for clinicians. 

Additionally, the swabs were performed in an outpatient clinic, as mentioned in the manuscript, not in an emergency department, and we aimed at investigating the symptoms presented at the time of the swab to observe if specific symptoms could be predictive of testing positivity, thus the follow up swab would have added, in our opinion, not significant information to the aim of the study. 

How can you exclude that you had a false positive result? For example, more than 200 children from the "negative" group had known contacts. Also, a number of children from the negative group had anosmia and ageusia, which you itself define as a predictive symptoms. Importantly, these symptoms are very unusual in other diseases.

As a proof that the study, although well written and analysed, has low significance, you define that a very common and usual presentation of children in the ED can be predictive of covid. However, we know that so many children can have mild or asymptomatic disease.

***Thank you for your comment. The sensitivity and specificity of the RT-PCR nasopharyngeal swab to detect SARS-CoV-2 declared by the manufacturer are very high (about 100%), so the possibility to have a false positive or negative can be considered pretty low, as also showed in the literature (1). 

Moreover, regarding the false positives, the swab and kits used for the analysis are certified to identify 3 different genes for all variants that are circulating to date. For each swab the quantity and quality of the material taken was checked, controlling that it was sufficient to highlight up to 5 copies of the virus (lower limit of the test). In case of a doubtful result (eg 2 out of 3 genes positive) the patient was asked to perform again the swab, thus the possibility to have false positives/negatives is very remote. This further information has been included in the manuscript in the section Laboratory methods.

About the lost among the cases tested negative possible positive SARS-CoV-2 might be realistic, we observed the trend on another study we conducted on serology, but the aim of this paper was not to demonstrate the circulation of the virus.

Furthermore, we do agree that the child can have been asymptomatic or mild, but firstly at that time of the pandemic all children also very slight symptoms were tested, and secondly this study explicitly included only children with symptoms to provide a predictive score to identify children at risk of COVID-19, we were not aiming at identifying all children with a positivity to SARS-CoV-2.

We also would like to thank you also for the comment on anosmia and ageusia. Although the symptoms are quite specific, those are not exclusive of COVID-19, but they might have other causes (such as allergies, other viral upper respiratory tract infections, seasonality) (2-3). Thus, those negatives presenting anosmia or ageusia might not be per se false negatives. Furthermore, also other studies reported anosmia and ageusia both in positive and negative cases, but they are more common among positives (3-5), this information has been inserted in the discussion. 

Finally, as we mentioned in our limits families were directly providing information, and some symptoms might have been misinterpreted, and anosmia and ageusia might be one of these considering the difficulty in detecting such symptoms in children. 

1. Floriano I, Silvinato A, Bernardo WM, Reis JC, Soledade G. Accuracy of the Polymerase Chain Reaction (PCR) test in the diagnosis of acute respiratory syndrome due to coronavirus: a systematic review and meta-analysis. Rev Assoc Med Bras (1992). 2020 Jul;66(7):880-888. doi: 10.1590/1806-9282.66.7.880. Epub 2020 Aug 24. PMID: 32844930.

2. Pierron D, Pereda-Loth V, Mantel M, Moranges M, Bignon E, Alva O, Kabous J, Heiske M, Pacalon J, David R, Dinnella C, Spinelli S, Monteleone E, Farruggia MC, Cooper KW, Sell EA, Thomas-Danguin T, Bakke AJ, Parma V, Hayes JE, Letellier T, Ferdenzi C, Golebiowski J, Bensafi M. Smell and taste changes are early indicators of the COVID-19 pandemic and political decision effectiveness. Nat Commun. 2020 Oct 14;11(1):5152. doi: 10.1038/s41467-020-18963-y. PMID: 33056983; PMCID: PMC7560893.

3. Haehner A, Draf J, Dräger S, de With K, Hummel T. Predictive Value of Sudden Olfactory Loss in the Diagnosis of COVID-19. ORL J Otorhinolaryngol Relat Spec. 2020;82(4):175-180. doi: 10.1159/000509143. Epub 2020 Jun 11. PMID: 32526759; PMCID: PMC7360503.

4. Sudre CH, Keshet A, Graham MS, Joshi AD, Shilo S, Rossman H, Murray B, Molteni E, Klaser K, Canas LD, Antonelli M, Nguyen LH, Drew DA, Modat M, Pujol JC, Ganesh S, Wolf J, Meir T, Chan AT, Steves CJ, Spector TD, Brownstein JS, Segal E, Ourselin S, Astley CM. Anosmia, ageusia, and other COVID-19-like symptoms in association with a positive SARS-CoV-2 test, across six national digital surveillance platforms: an observational study. Lancet Digit Health. 2021 Sep;3(9):e577-e586. doi: 10.1016/S2589-7500(21)00115-1. Epub 2021 Jul 22. Erratum in: Lancet Digit Health. 2021 Sep;3(9):e542. PMID: 34305035; PMCID: PMC8297994.

5. Mak PQ, Chung KS, Wong JS, Shek CC, Kwan MY. Anosmia and Ageusia: Not an Uncommon Presentation of COVID-19 Infection in Children and Adolescents. Pediatr Infect Dis J. 2020 Aug;39(8):e199-e200. doi: 10.1097/INF.0000000000002718. PMID: 32516281.

I am very very sorry to see that you did a huge effort in this analyses, but really the hypothesis behind the design of this study has a very doubtful significance. You should use these data to make a new one, doing f-up swabs as a minimum, or interview of parents to understand if further swabs have been done and the results to see how many of them are true negative. For example, a preprint from UK recently found that 1 out of 3 students had IgG positive (!!) confirming that the virus circulates much more than expected. And I am quite sure that among the negative, particularly those with known contacts and with anosmia/ageusia, you missed several cases.

***Thank you for your comment. We acknowledge the fact that the high rate of asymptomatic children might limit the predictiveness of symptoms, but we believe that this is an independent variable from the false positive issue. 

Moreover, IgG testing measures positivity in the span of several months, so that in a methodologic perspective, could not be related to symptoms or swab positivity in a defined moment. Finally, as mentioned above, the aim of this paper was not to demonstrate the circulation of the virus, but to identify clinical, anamnestic, and sociodemographic characteristics associated with a positive swab for SARS-CoV-2 in a population of children.

Reviewer #2: In the manuscript authored by Benedetta Armocida et al, authors summarize clinical, anamnestic, and sociodemographic characteristics of children affected or not by Sars Cov-2 in a frame period between September 20th and December 23rd 2020.

A major concern is that authors provided a study that is more descriptive then predictive, as stated from authors. This is a critical point. 

***Thank you for your comment. We acknowledge the fact that the high rate of asymptomatic children makes limits the predictiveness of symptoms, but we believe that this is an independent variable from the false positive issue. 

We agree that in a blurred clinical setting, as SARS-CoV-2 infection in children, every study based on symptoms carries the risk of being more descriptive than predictive. This matter has been specified in text as a limitation, although concrete data on these issues are scant. 

No follow up is present. 

***Thank you for your comment. At that time the general process after a positive swab envisaged the activation of quarantine and a swab at two weeks by the department of prevention. Thus, a follow up (control) swab was always performed, but these follow up swabs could have been performed in other departments or hospitals, not necessary in our Institute. Moreover, the aim of this study was to identify clinical, anamnestic, and sociodemographic characteristics associated with a positive swab for SARS-CoV-2in a population of children, hence we aimed at investigating a real life, pragmatic approach in order to try to add some clinical hints for clinicians and the symptoms presented at the time of the swab to observe if specific symptoms could be predictive of testing positivity, thus the follow up swab would have added, in our opinion, not significant information to the aim of the study. 

ROC curve is computed between 127 positive vs 1357 negative. Are these groups comparable?

***Thank for your comment. The two groups are comparable. As mentioned in the study population and in the inclusion criteria, all cases were children (2-14 years of age), coming from the general population of Trieste, all attending school or education services, and turning to the territorial services to undergo a RT-PCR nasopharyngeal swab to detect SARS-CoV-2, and only the ones with symptoms were included in the study. 

It is not clear to the reader if authors suggest NOT to perform Sars-Cov2 diagnostic test in children presenting rhinorrhoea, sore throat, or diarrhoea.

***Thank for your comment. To clarify, we are not suggesting clinicians not to perform SARS-CoV-2 diagnostic test in children presenting rhinorrhea, sore throat, or diarrhea, although we add the information that these symptoms carry a low risk of SARS-COV 2 infection. The information might be very relevant and might be helpful in orienting choices in specific settings, such as these with limited resources. This information has been included in the Discussion

---

## [Decision Letter · Decision Letter 1]

3 Nov 2021

PONE-D-21-24181R1Clinical, anamnestic, and sociodemographic predictors of positive SARS-CoV-2 testing in children: a cross sectional study in a tertiary hospital in ItalyPLOS ONE

Dear Dr. Ronfani,

Thank you for submitting your manuscript to PLOS ONE. After careful consideration, we feel that it has merit but does not fully meet PLOS ONE’s publication criteria as it currently stands. Therefore, we invite you to submit a revised version of the manuscript that addresses the points raised during the review process.

Please revise.

We look forward to receiving your revised manuscript.

Kind regards,

Academic Editor

PLOS ONE

Reviewers' comments:

Reviewer's Responses to Questions

**Comments to the Author**

1. If the authors have adequately addressed your comments raised in a previous round of review and you feel that this manuscript is now acceptable for publication, you may indicate that here to bypass the “Comments to the Author” section, enter your conflict of interest statement in the “Confidential to Editor” section, and submit your "Accept" recommendation.

Reviewer #1: (No Response)

Reviewer #2: All comments have been addressed

2. Is the manuscript technically sound, and do the data support the conclusions?

Reviewer #1: No

Reviewer #2: Yes

3. Has the statistical analysis been performed appropriately and rigorously? 

Reviewer #1: No

Reviewer #2: Yes

4. Have the authors made all data underlying the findings in their manuscript fully available?

Reviewer #1: No

Reviewer #2: Yes

5. Is the manuscript presented in an intelligible fashion and written in standard English?

Reviewer #1: Yes

Reviewer #2: Yes

6. Review Comments to the Author

Reviewer #1: I still think that a very single test cannot fully exclude covid in the negative control group, and therefore a study aiming to detect positive cases at the time of testing, without knowing days of illness and further follow-up is of limited value for a prestigious journal

Reviewer #2: Authors provided most of the corrections requested from the reviewers and now the manuscript has been higly improved.

7. PLOS authors have the option to publish the peer review history of their article (what does this mean?). If published, this will include your full peer review and any attached files.

Reviewer #1: No

Reviewer #2: No

---

## [Author Response · Author response to Decision Letter 1]

2 Dec 2021

Review Comments to the Author

Reviewer #1: I still think that a very single test cannot fully exclude covid in the negative control group, and therefore a study aiming to detect positive cases at the time of testing, without knowing days of illness and further follow-up is of limited value for a prestigious journal

***Thank you for your comment. We acknowledge and agree that a single test cannot completely exclude the positives in the negatives control group, therefore we explicitly included this point as limitations of the study. However, we considered this possibility extremely small, as our Institute was the only hub performing COVID-19 swab in the Trieste area (information added at the beginning of the methods section), hence in case of persistent or worsening symptoms, the access of these children for a second swab would have been performed within our Institute and this didn’t happen from our available data. 

Reviewer #2: Authors provided most of the corrections requested from the reviewers and now the manuscript has been higly improved.

***Thank you for your appreciation.

---

## [Decision Letter · Decision Letter 2]

21 Dec 2021

PONE-D-21-24181R2Clinical, anamnestic, and sociodemographic predictors of positive SARS-CoV-2 testing in children: a cross sectional study in a tertiary hospital in ItalyPLOS ONE

Dear Dr. Ronfani,

Thank you for submitting your manuscript to PLOS ONE. After careful consideration, we feel that it has merit but does not fully meet PLOS ONE’s publication criteria as it currently stands. Therefore, we invite you to submit a revised version of the manuscript that addresses the points raised during the review process.

Please revise.

We look forward to receiving your revised manuscript.

Kind regards,

Academic Editor

PLOS ONE

Journal Requirements:

Reviewers' comments:

Reviewer's Responses to Questions

**Comments to the Author**

1. If the authors have adequately addressed your comments raised in a previous round of review and you feel that this manuscript is now acceptable for publication, you may indicate that here to bypass the “Comments to the Author” section, enter your conflict of interest statement in the “Confidential to Editor” section, and submit your "Accept" recommendation.

Reviewer #1: All comments have been addressed

Reviewer #3: (No Response)

2. Is the manuscript technically sound, and do the data support the conclusions?

Reviewer #1: Yes

Reviewer #3: Yes

3. Has the statistical analysis been performed appropriately and rigorously? 

Reviewer #1: Yes

Reviewer #3: Yes

4. Have the authors made all data underlying the findings in their manuscript fully available?

Reviewer #1: No

Reviewer #3: Yes

5. Is the manuscript presented in an intelligible fashion and written in standard English?

Reviewer #1: Yes

Reviewer #3: Yes

6. Review Comments to the Author

Reviewer #1: thank you very much for this clarification, and thank you for having included it in the limitation section. this is an helpful addition and I think also you will benefit from this clarification when other colleagues will read this paper.

I only ask one little addition:

the authors NEVER compared in the discussion their findings with other italian ones, this should be done and particularly with an early national italian report , to also address if symptoms presentation have changed after the very early months (please refer to COVID-19 in 17 Italian Pediatric Emergency Departments, Parri, N., et al, Pediatrics, 2020, 146(6), e20201235) and even other ones would be appreciated

Reviewer #3: This is an interesting study performed by Armocida et al, investigating the clinical, anamnestic and sociodemographic factors at the time of a positive SARS-CoV-2 test in children. The authors constructed a predictive model of SARS-CoV-2 positivity.

My only concern is that whether any of the negative patients subsequently developed SARS-CoV-2. The authors state that none of the negative patients tested positive within their laboratory. Do they have data on whether children were retested and gave a negative test or there simply were no positive tests? My suspicion would be that if a child was negative then the likelihood of being retested would be relatively low. Without systematically testing patients while they continued to have symptoms then it is difficult to definitively answer this question. The authors must make it clear that these are the symptoms that the children had whilst they tested positive.

In addition, do the authors have any data on how long the symptoms persisted prior to a positive test?

7. PLOS authors have the option to publish the peer review history of their article (what does this mean?). If published, this will include your full peer review and any attached files.

Reviewer #1: No

Reviewer #3: No

---

## [Author Response · Author response to Decision Letter 2]

30 Dec 2021

PONE-D-21-24181R2

Clinical, anamnestic, and sociodemographic predictors of positive SARS-CoV-2 testing in children: a cross sectional study in a tertiary hospital in Italy

PLOS ONE

Reviewer #1: thank you very much for this clarification, and thank you for having included it in the limitation section. this is an helpful addition and I think also you will benefit from this clarification when other colleagues will read this paper.

I only ask one little addition:

the authors NEVER compared in the discussion their findings with other italian ones, this should be done and particularly with an early national italian report , to also address if symptoms presentation have changed after the very early months (please refer to COVID-19 in 17 Italian Pediatric Emergency Departments, Parri, N., et al, Pediatrics, 2020, 146(6), e20201235) and even other ones would be appreciated

***Thank you very much for this additional request. We have included two Italian studies as references and compared the main findings also with an Italian study carried out in a similar outpatient setting and in the same school aged population (reference 8).

References added: 

Parri N, Lenge M, Cantoni B, Arrighini A, Romanengo M, Urbino A, et al. COVID-19 in 17 Italian Pediatric Emergency Departments. Pediatrics. 2020 Dec;146(6):e20201235. doi: 10.1542/peds.2020-1235. Epub 2020 Sep 23. PMID: 32968031.

Lazzerini M, Sforzi I, Trapani S, Biban P, Silvagni D, Villa G, et al. COVID-19 Italian Pediatric Study Network. Characteristics and risk factors for SARS-CoV-2 in children tested in the early phase of the pandemic: a cross-sectional study, Italy, 23 February to 24 May 2020. Euro Surveill. 2021 Apr;26(14):2001248. doi: 10.2807/1560-7917.ES.2021.26.14.2001248. PMID: 33834960; PMCID: PMC8034058. 

As mentioned in the first paragraph of our discussion though, to our knowledge, this is the first study to provide a predictive score for the paediatric population to identify children at risk to be positive to the SARS-CoV-2 test. 

Reviewer #3: This is an interesting study performed by Armocida et al, investigating the clinical, anamnestic and sociodemographic factors at the time of a positive SARS-CoV-2 test in children. The authors constructed a predictive model of SARS-CoV-2 positivity.

My only concern is that whether any of the negative patients subsequently developed SARS-CoV-2. The authors state that none of the negative patients tested positive within their laboratory. Do they have data on whether children were retested and gave a negative test or there simply were no positive tests? My suspicion would be that if a child was negative then the likelihood of being retested would be relatively low. Without systematically testing patients while they continued to have symptoms then it is difficult to definitively answer this question. The authors must make it clear that these are the symptoms that the children had whilst they tested positive.

***Thank you very much for this comment. Unfortunately, we don’t have this additional information. As we mentioned in the limits, we acknowledge the possible limit in performing only one single test, which, without a further systematic follow-up, might have not detected a child positive to SARS-CoV-2, who might have been included in the negative control group. However, we considered this possibility extremely small, as our Institute was the only hub performing COVID-19 swab in the Trieste area, hence in case of persistent or worsening symptoms, the access of these children for a second swab would have been performed within our Institute and this didn’t happen from our available data. 

We added a sentence in the Methods section to clarify that we collected information on specific symptoms that led to the swab being administered.

In addition, do the authors have any data on how long the symptoms persisted prior to a positive test?

***Thank you very much for this comment. Unfortunately, we don’t have this additional information as data were collected at the time the RT-PCR testing was performed. This information was reported in Methods “data collection” section.

---

## [Decision Letter · Decision Letter 3]

10 Jan 2022

Clinical, anamnestic, and sociodemographic predictors of positive SARS-CoV-2 testing in children: a cross sectional study in a tertiary hospital in Italy

PONE-D-21-24181R3

Dear Dr. Ronfani,

We’re pleased to inform you that your manuscript has been judged scientifically suitable for publication and will be formally accepted for publication once it meets all outstanding technical requirements.

Kind regards,

Academic Editor

PLOS ONE

Additional Editor Comments (optional):

Reviewers' comments:

Reviewer's Responses to Questions

**Comments to the Author**

1. If the authors have adequately addressed your comments raised in a previous round of review and you feel that this manuscript is now acceptable for publication, you may indicate that here to bypass the “Comments to the Author” section, enter your conflict of interest statement in the “Confidential to Editor” section, and submit your "Accept" recommendation.

Reviewer #1: All comments have been addressed

Reviewer #3: All comments have been addressed

2. Is the manuscript technically sound, and do the data support the conclusions?

Reviewer #1: Yes

Reviewer #3: Yes

3. Has the statistical analysis been performed appropriately and rigorously? 

Reviewer #1: Yes

Reviewer #3: Yes

4. Have the authors made all data underlying the findings in their manuscript fully available?

Reviewer #1: No

Reviewer #3: Yes

5. Is the manuscript presented in an intelligible fashion and written in standard English?

Reviewer #1: Yes

Reviewer #3: Yes

6. Review Comments to the Author

Reviewer #1: despite several limitations, the authors have replied to all concerns raised by authors.

good luck.

despite several limitations, the authors have replied to all concerns raised by authors.

good luck.

Reviewer #3: (No Response)

7. PLOS authors have the option to publish the peer review history of their article (what does this mean?). If published, this will include your full peer review and any attached files.

Reviewer #1: No

Reviewer #3: **Yes: **Dr Simon T Abrams

---

## [Editor Report · Acceptance letter]

14 Jan 2022

PONE-D-21-24181R3 

Clinical, anamnestic, and sociodemographic predictors of positive SARS-CoV-2 testing in children: a cross sectional study in a tertiary hospital in Italy 

Dear Dr. Ronfani:

I'm pleased to inform you that your manuscript has been deemed suitable for publication in PLOS ONE. Congratulations! Your manuscript is now with our production department. 

Kind regards, 

on behalf of

Dr. Robert Jeenchen Chen 

Academic Editor

PLOS ONE